# Span-Pair Interaction and Tagging for Dialogue-Level Aspect-Based Sentiment Quadruple Analysis

Submission Id: 259

## ABSTRACT

The Dialogue-level Aspect-based Sentiment Quadruple analysis (DiaASQ) task has recently received attention in the Aspect-Based Sentiment Analysis (ABSA) field. It aims to extract *(target, aspect, opinion, sentiment)* quadruples from multi-turn and multi-party dialogues. Compared to previous ABSA tasks focusing on text such as sentences, the DiaASQ task involves more complex contextual information and corresponding relations between terms, as well as longer sequences. These characteristics challenge existing methods that struggle to model explicit span-level interactions or have high computational costs. In this paper, we propose a span-pair interaction and tagging method to solve these issues, which includes a novel Span-pair Tagging Scheme (STS) and a simple and efficient Multi-level Representation Model (MRM). STS simplifies the DiaASQ task to a span-pair tagging task and explicitly captures complete span-level semantics by tagging span pairs. MRM efficiently models the dialogue structure information and span-level interactions by constructing multi-level contextual representation. Besides, we train a span ranker to improve the running efficiency of MRM. Extensive experiments on multilingual datasets demonstrate that our method outperforms existing state-of-the-art methods.

## CCS CONCEPTS

• **Information systems** → **Sentiment analysis**; • **Computing methodologies** → **Natural language processing**.

## KEYWORDS

Natural language processing, Aspect-based sentiment analysis, Quadruple extraction, Dialogue scene

**ACM Reference Format:**

Anonymous Author(s). 2018. Span-Pair Interaction and Tagging for Dialogue-Level Aspect-Based Sentiment Quadruple Analysis. In *Proceedings of Make sure to enter the correct conference title from your rights confirmation emai (Conference acronym 'XX).* ACM, New York, NY, USA, 11 pages. https://doi.org/XXXXXXX.XXXXXXX

## 1 INTRODUCTION

Aspect-Based Sentiment Analysis (ABSA) is a crucial research field in natural language processing, which aims to determine the sentiment polarities towards specific aspects of targets [18, 23]. ABSA

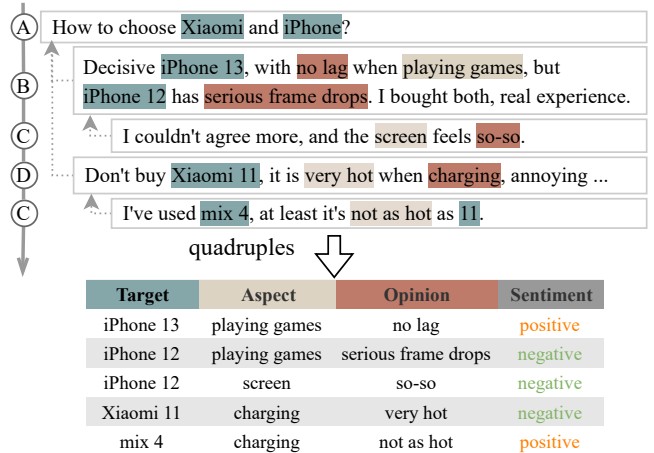

**Figure 1: An example of a DiaASQ task. The capital letters A, B, C, and D identify the speakers. Dotted lines with arrows show the reply relations between utterances. The quadruple consists of three terms (i.e., *target, aspect, opinion*) and a sentiment polarity.**

has found wide applications in E-commerce platforms, where it helps improve products and services based on customer feedback from web content.[1] In recent years, there has been a growing trend of discussing products and services through multi-turn and multi-party dialogues on social media platforms such as Twitter and Weibo. However, existing ABSA methods [2, 5, 16, 17, 19, 20, 22, 29, 35] primarily focus on text-level ABSA tasks, analyzing individual texts such as sentences and documents while ignoring more complex and dynamic dialogue scenes.

To address this limitation and promote the development of dialogue-level ABSA, Li et al. [14] proposed the Dialogue-level Aspect-based Sentiment Quadruple analysis (DiaASQ) task. This task aims to extract *(target, aspect, opinion, sentiment)* quadruples from multi-turn and multi-party dialogues, as illustrated in Figure 1. Compared to text-level ABSA tasks, DiaASQ exhibits the following characteristics: 1) More complex contextual information resulting from the structure of multi-turn and multi-party dialogues. 2) More complex corresponding relations between terms resulting from more types of terms (a new term *target*). 3) Longer sequences due to the inclusion of multiple utterances in dialogue. These characteristics challenge existing methods in addressing the DiaASQ task.

---

[1] **Relevance to the Web**: ABSA belongs to *Sentiment analysis and opinion mining* Topic of *Web Mining and Content Analysis* Track, which aims to analyze and extract structural sentiment information from web content. The works involved in this paper aim to make web content more harmless and helpful for online shopping and communication.

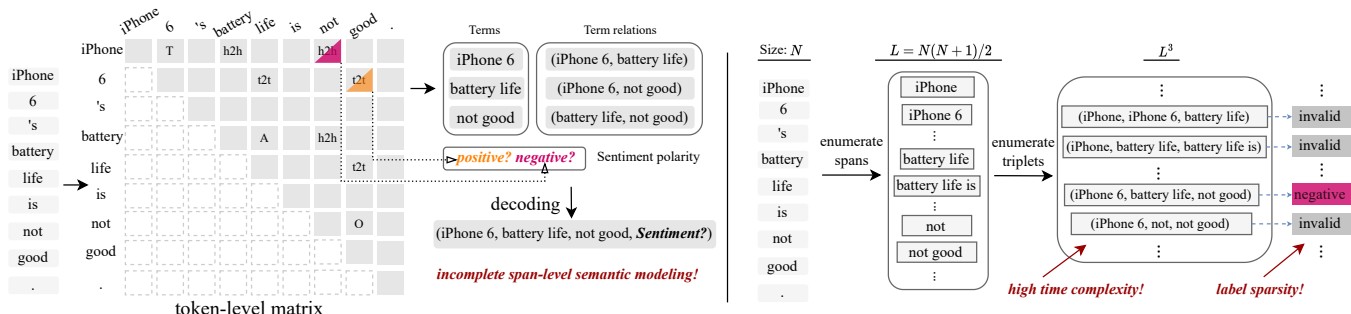

**Figure 2:** *Left*: Token-pair tagging method proposed by Li et al. [14]. The T/A/O label denotes the target/aspect/opinion term. The h2h and t2t identify the relation between spans by aligning the head and tail tokens between spans. The sentiment polarity label is attached to the h2h and t2t labels between the target and opinion terms. *Right*: Span-level enumeration method.

Token-pair tagging is a popular method for text-level ABSA tasks [4, 17, 26, 34], and Li et al. [14] extended the classical token-pair tagging method GTS [26] for the DiaASQ task. As illustrated on the left side of Figure 2, given a dialogue snippet containing $N$ tokens, the method first extracts terms, relations, and sentiment polarities by tagging each token pair in an $N \times N$ token-level matrix and then decodes quadruples from the extracted results. **However, the token-pair tagging method cannot capture complete semantic information of spans due to the lack of span-level modeling and interactions.** For example, this method identifies the token pair *(iPhone, not)* with *negative* sentiment and the *(6, good)* with *positive* sentiment, leading to failure to determine the sentiment polarity of the quadruple. The complex contextual information and corresponding relations between terms in the DiaASQ task exacerbate the issue. To this end, some works [6, 10, 28] proposed span-level enumeration methods to capture the complete span-level semantic information. **However, this method leads to high computational costs and label sparsity problems due to the more types of terms and longer sequences involved in the DiaASQ task.**. As shown on the right side of Figure 2, this method enumerates spans to create a span list of size $L = N(N + 1)/2$. Then, it arranges spans into *(target, aspect, opinion)* triplets and predicts their sentiment polarities, leading to a high time complexity ($O(L^3)$). Besides, the abundant invalid triplets cause label sparsity problems that hinder training from reaching optimal performance.

In this paper, we propose a span-pair interaction and tagging method, comprising a novel Span-pair Tagging Scheme (STS) and a simple and efficient Multi-level Representation Model (MRM), to solve the issues of the above two kinds of methods. In contrast to previous works of tagging token pairs, STS can explicitly capture complete span-level semantic information by tagging span pairs in the span-level matrix. Meanwhile, MRM generates the $L \times L$ span-level matrix by modeling the multi-level information. Compared to the span-level enumeration method, MRM exhibits stable time complexity at $O(L^2)$ and does not increase with the number of term types.

Concretely, STS extracts target, aspect, and opinion terms by tagging the span-level matrix diagonal. It also extracts the type of relations between terms and sentiment polarities by tagging the strictly upper triangular region of the matrix. Then, STS decodes

quadruples under the verification of terms and the type of relations between terms. Furthermore, MRM models the dialogue structure information at the token level by self-attention mechanism [24] and models interactions between spans at the span level by Hadamard product operation, respectively. It enumerates all spans and constructs a $L \times L$ span-level matrix. Then, it obtains quadruples with the help of STS. The simple structure of MRM ensures the $O(L^2)$ time complexity. Besides, we train a span ranker to improve the running efficiency of MRM. It further optimizes the time complexity to $O(K^2)$ by selecting the top $K$ spans to reduce the size of the matrix from $L \times L$ to $K \times K$ ($K \ll L$, $K < N$, $N$ is the number of tokens, and $L$ is the number of spans). Our contribution can be summarized as follows:

(1) We design a Span-pair Tagging Scheme (STS) that explicitly capture complete span-level semantic information by tagging span pairs rather than token pairs. To our best knowledge, this is the first time to solve the dialogue-level ABSA task by tagging span pairs.

(2) We propose a simple and efficient Multi-level Representation Model (MRM), which explicitly models the dialogue information and the interactions between spans at the token and span level, respectively. Besides, we train a span ranker to make MRM run more efficiently than previous models.

(3) We conduct extensive experiments on multilingual DiaASQ datasets. The experimental results demonstrate that our method outperforms the state-of-the-art methods.[2]

## 2 RELATED WORK

### 2.1 Text-level Aspect-Based Sentiment Analysis

Text-level Aspect-Based Sentiment Analysis (ABSA) is a fine-grained sentiment analysis problem that aims to determine the opinion and sentiment polarity at the aspect level from sentences or documents [12, 18, 23, 33, 36]. Early works focused on single ABSA tasks such as Aspect Term Extraction (ATE) [19, 25] and Aspect Sentiment Classification (ASC) [8, 15, 27]. However, recent studies have explored compound ABSA tasks such as Aspect-Opinion Pair

---

[2]The code and datasets is located in Anonymous GitHub: https://anonymous.4open.science/r/WWW24-ID259.

Extraction (AOPE) [11, 26, 35], Aspect Sentiment Triplet Extraction (ASTE) [13, 26, 28, 34], and Aspect Sentiment Quad Prediction (ASQP) [3, 30] due to their practicality. Next, we will focus on the ASTE task and provide a detailed explanation of its methods as it is most relevant to the DiaASQ task. ASTE is a prevalent compound ABSA task that aims to extract *(aspect, opinion, sentiment)* triplets from texts such as sentences [22]. Wu et al. [26] proposed the Grid Tagging Scheme (GTS) for the ASTE task. GTS extracts aspect terms, opinion terms, and sentiment polarities by tagging token pairs. To alleviate the boundary insensitivity and relation inconsistency problems of GTS, Zhang et al. [34] and Liang et al. [17] proposed Boundary-Driven Table-Filling (BDTF) and Span TAgging and Greed infErence (STAGE) methods, respectively. However, these methods cannot explicitly capture complete span-level semantic information because they all belong to the token-pair tagging scheme. To this end, span-level enumeration methods are proposed [6, 10, 28]. This method enumerates all spans, arranges them into *(aspect, opinion)* pairs, and predicts their sentiment polarities. Xu et al. [28] proposed an end-to-end model, Span-ASTE, which includes a dual-channel span pruning strategy to ease the high computational cost. Chen et al. [6] proposed a span-level bidirectional network to extract triplets from both aspect-to-opinion and opinion-to-aspect directions. Feng et al. [10] infused syntax knowledge into the span-level enumeration method. However, the time complexity of the span-level enumeration methods increases exponentially with the increase of the term types.

## 2.2 Dialogue-level Aspect-based Sentiment Quadruple Analysis

In recent years, there has been a proliferation of conversational scenes on social media platforms. Individuals increasingly use posts and comments on sites such as Twitter and Weibo to discuss products and services in a dialogue format. In order to promote the development of ABSA in dialogue scenarios, Li et al. [14] proposed the Dialogue-level Aspect-based Sentiment Quadruple analysis (DiaASQ) task and annotated large-scale datasets in both Chinese and English. The DiaASQ task aims to extract *(target, aspect, opinion, sentiment)* quadruples given multi-turn and multi-party dialogues. Compared with text-level ABSA, DiaASQ adds the new term *target*, which denotes words or phrases that refer to the evaluated object, such as *iPhone 6*. Li et al. [14] proposed an end-to-end token-pair tagging method based on GTS [26]. The method models the dialogue information by a multi-head attention mechanism [24] and designs a new token-pair tagging scheme. However, it still cannot explicitly capture the complete span-level semantic information. Besides, Li et al. [14] evaluated the span-level enumeration method on the DiaASQ task. However, high computational costs and label sparsity problems caused by characteristics of the DiaASQ task degrade its performance and efficiency. Before the span pair-level method proposed in this paper, no work could still efficiently model full span-level semantics in the DiaASQ task.

## 3 METHODOLOGY

### 3.1 Task Formulation

A dialogue contains multiple utterances $D = (u_1, u_2, \cdots, u_{|D|})$ with the corresponding reply sequence $Re = (re_1, re_2, \cdots, re_{|D|})$ and

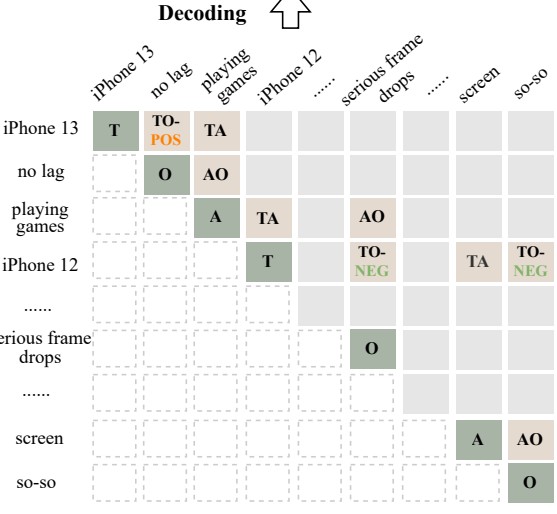

Figure 3: An example of span-pair tagging. Due to the space constraints, we selected a few candidate spans from Figure 1 to form the span-level matrix.

speaker sequence $Sp = (sp_1, sp_2, \cdots, sp_{|D|})$, where $re_i$ denotes the index of the utterance that the $i$-th utterance replies to and $sp_i$ denotes the speaker identity of the $i$-th utterance[3]. The goal of the DiaASQ task is to extract all *(target, aspect, opinion, sentiment)* quadruples in $D$. The quadruple set is denoted as:

$$Q = \{(t_k, a_k, o_k, p_k)\}_{k=1}^{|Q|}, \tag{1}$$

where $t$, $a$, and $o$ are spans from origin dialogue $D$ and represent target, aspect, and opinion term, respectively. $p \in \{positive, negative, other\}$ is sentiment polarity.

### 3.2 Span-pair Tagging Scheme

In this section, we first introduce the definition of span-pair labels. Then, we provide a straightforward decoding process. Finally, we summarize several differences between the Span-pair Tagging Scheme (STS) and the token-pair tagging scheme proposed by Li et al. [14] to further emphasize the advantages of STS.

*3.2.1 The Definition of Span-pair Labels.* 1) Term Types: We use the T, A, and O labels to denote the target, aspect, and opinion terms. For example, the *screen* is an aspect term if the span pair *(screen, screen)* is tagged with A. 2) Relations between Terms: We use TA, AO, and TO labels to denote the three relations between terms, namely target-aspect relation, aspect-opinion relation, and target-opinion relation. For example, if the span pair *(iPhone 12, screen)* is tagged with TA, the relation between *iPhone 12* and *screen* is a target-aspect relation. 3) Sentiment Polarities: We use POS, NEG, and OTH labels to denote the positive, negative, and other sentiment polarities. To ensure consistency in the tagging scheme,

---

[3]The $|*|$ denotes the number of elements in the collection $*$.

we attach sentiment labels to the TO label to form TO-POS, TO-NEG, and TO-OTH labels. For example, there is a target-opinion relation with positive sentiment if the span pair *(iPhone 13, no lag)* is tagged with TO-POS. All span-pair labels are as follows:

$$\{T, A, O, TA, AO, \text{TO-POS}, \text{TO-NEG}, \text{TO-OTH}, \text{None}\}. \quad (2)$$

Figure 3 shows a matrix tagged by span-pair labels. It is worth noting that labels {T, A, O} are only in the matrix diagonal, while labels {TA, AO, TO-POS, TO-NEG, TO-OTH} are only in the strictly upper triangular region of the matrix.

*3.2.2 The Decoding Process.* The design of span-pair labels makes the decoding process more straightforward and intuitive. When decoding quadruples, we first obtain the *(target, aspect, opinion)* triplets under the supervision of labels {T, A, O} and {TA, AO}. Then, we filter invalid triplets and identify sentiment polarities by labels {TO-POS, TO-NEG, TO-OTH}. The detailed decoding process and algorithm are in the Appendix A.

*3.2.3 Differences from Token-pair Tagging Scheme.* Li et al. [14] first applied the token-pair tagging scheme to the DiaASQ task by retrofitting the GTS [26]. For convenience, we refer to the token-pair tagging scheme proposed by Li et al. [14] as the GTS-Dia scheme. Our STS differs from the GTS-Dia scheme in the following ways:

(1) **The granularity of tagging is different**. STS tags at the span level, while GTS-Dia scheme at the token level. Therefore, STS can explicitly capture the complete semantics of spans and model the span-level interactions while the GTS-Dia scheme fails.

(2) **The difficulty of tagging is different.** The labels {T, A, O} and labels {TA, AO, TO-POS, TO-NEG, TO-OTH} are distributed across different regions of the span-level matrix in the STS, which reduces the difficulty of tagging. In other words, the model can narrow down the choice of labels by identifying whether two spans in a span pair are the same or not. In contrast, in the GTS-Dia scheme, all labels will likely appear in the strictly upper triangular region of the token-level matrix, making tagging more challenging.

(3) **The difficulty and efficiency of decoding are different.** The decoding process of STS is both straightforward and intuitive thanks to the design of span-pair labels, as mentioned in section 3.2.2. However, the GTS-Dia scheme requires a human-designed heuristic algorithm for complicated situations. One such issue is the difficulty in determining the correspondence between h2h and t2t labels. Besides, it can be tricky to determine the sentiment polarity when the sentiment labels attached to h2h and t2t are different. These issues complicate the decoding process of the GTS-Dia scheme.

## 3.3 Multi-level Representation Model

Figure 4 shows an overview of our Multi-level Representation Model (MRM).

**Token Encoding**: We use Pre-trained Language Models (PLMs) as the contextual encoder of the dialogue $D = (u_1, u_2, \cdots, u_{|D|})$. In order to take full advantage of the capabilities of the PLMs, the whole dialogue $D$ with $< s >$ and $< /s >$ is fed into the PLMs:

$$I = (< s >, u_1, < /s >, \cdots, < s >, u_{|D|}, < /s >), \quad (3)$$

$$H = (h_1, h_2, \cdots, h_{|I|}) = \text{PLMs}(I), \quad (4)$$

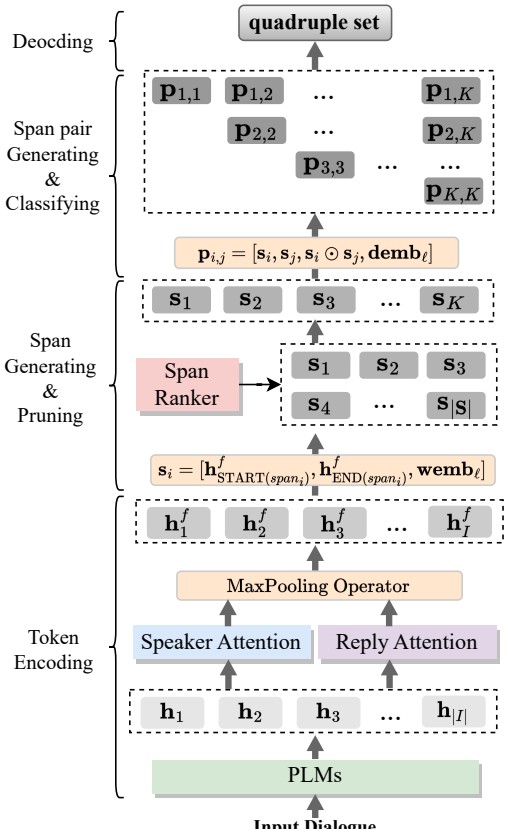

**Figure 4: Multi-level Representation Model.**

where $< s >$ and $< /s >$ denote special tokens of PLMs, $h_i$ denotes the contextual representation of the $i$-th token of input sequence $I$. We construct the reply mask $M^{Re}$ and speaker mask $M^{Sp}$ to denote the reply sequence $Re$ and speaker sequence $Sp$, respectively. Following previous work [14], we use multi-head self-attention mechanisms [24] to infuse this information:

$$H^* = \text{Masked-MultiHead-Attn}\left(Q, K, V, M^*\right)$$
$$= \text{softmax}\left(\frac{(Q^T K) \odot M^*}{\sqrt{d}}\right) V, \quad (5)$$

where $* \in \{Re, Sp\}$, $Q = K = V = H$, $\odot$ denotes Hadamard product operation, and $d$ denotes the hidden size. Next, we use the MaxPooling operation to obtain the contextual representations of tokens with dialogue structure information:

$$H^f = (h_1^f, h_2^f, \cdots, h_{|I|}^f) = \text{MaxPooling}(H^{Re}, H^{Sp}). \quad (6)$$

**Span Generating & Pruning**: We obtain the span list $SL$ by enumerating all spans. Then, we define the contextual representation of each span by infusing boundary and width information and obtain the span's contextual representation list $S$ corresponding to $SL$:

$$s_i = [h_{\text{START}(span_i)}^f, h_{\text{END}(span_i)}^f, \text{wemb}_\ell],$$
$$S = (s_1, s_2, ..., s_L), \quad (7)$$

where $span_i \in SL$ denotes the $i$-th span, $L = |SL| = |S|$ denotes the total number of spans, $\text{START}(span_i)$ and $\text{END}(span_i)$ denotes the index of the start and end token for the $i$-th span, and $\mathbf{wemb}_\ell$ denotes the learnable embeddings of width $\ell = |\text{END}(span_i) - \text{START}(span_i) + 1|$. In order to improve running efficiency, we train a Span Ranker (It will be described below) to prune $SL$ and $\mathbf{S}$. The Span Ranker gives the score of term types for $span_i$:

$$score_i = P(\text{T}|span_i) + P(\text{A}|span_i) + P(\text{O}|span_i), \qquad (8)$$

where $P(\text{T}|span_i)$ denotes the probability that the $i$-th span is predicted as the target term. Then, we use the contextual representations of the top $K$ spans with the highest score to form a new contextual representation list:

$$\mathbf{S}_{top} = \{\mathbf{s}_1, \mathbf{s}_2, \cdots, \mathbf{s}_K\}, \qquad (9)$$

where $|\mathbf{S}_{top}| = K \ll L$.

**Span pair Generating & Classifying** To model the span-level interactions and the distance information between spans, we construct the contextual representation of each span pair as:

$$\mathbf{p}_{i,j} = [\mathbf{s}_i, \mathbf{s}_j, \mathbf{s}_i \odot \mathbf{s}_j, \mathbf{demb}_\ell], \qquad (10)$$

where $\mathbf{demb}_\ell$ denotes the learnable embeddings of distance $\ell = min(|\text{END}(span_j) - \text{START}(span_i)|, |\text{START}(span_j) - \text{END}(span_i)|)$, $1 \leq i \leq j \leq K$ and $\mathbf{s}_i, \mathbf{s}_j \in \mathbf{S}_{top}$. The contextual representations of all span pairs form the upper triangular matrix of size $K \times K$. We apply a Multi-Layer Perception (MLP) to predict the probability distribution of labels:

$$P(y_{i,j}) = \text{softmax}(\text{MLP}(\mathbf{p}_{i,j})), \qquad (11)$$

where the label $y_{i,j}$ is distributed among nine categories, as shown in Formula 2.

**Decoding** Finally, we obtain all quadruples from the tagged span-level matrix with the help of STS.

## 3.4 Span Ranker

We train a Part-Of-Speech-aware (POS-aware) Span Ranker to score candidate spans within the MRM. Like the MRM, we concatenate utterances by special tokens of PLMs and feed them into PLMs to obtain the initial contextual representation of tokens $\mathbf{H} = (\mathbf{h}_1, \mathbf{h}_2, \cdots, \mathbf{h}_{|I|})$, as shown in Formulas 3 and 4.

In fact, the type of term is closely related to its POS. For example, the *target* term is generally a proper noun or noun phrase, and the *opinion* term is generally an adjective or adjective phrase. Therefore, we enumerate all spans and obtain their boundary POS information using natural language processing tools (NLTK[4] for English and Jieba[5] for Chinese). Besides, we use the AveragePooling operation to obtain the overall semantic representation of each span. The final contextual representation of each span is as follows:

$$\mathbf{s}_{a,b}^{sr} = [\mathbf{h}_a, \mathbf{pemb}_{pos_a}, \mathbf{h}_b, \mathbf{pemb}_{pos_b}, \mathbf{wemb}_{|b-a+1|},$$
$$\text{AveragePooling}(\mathbf{h}_a, \mathbf{h}_{a+1}, \cdots, \mathbf{h}_b)], \qquad (12)$$

where superscript $sr$ is the abbreviation for Span Ranker, $\mathbf{pemb}_{pos}$ denotes the learnable embedding of POS $pos$, and $pos_a$ denotes the

---

[4]https://www.nltk.org/
[5]https://github.com/fxsjy/jieba

---

**Table 1: Statistics of DiaASQ dataset. The "Intra" denotes the intra-utterance quadruples, where terms come from the same utterances. The "Inter" denotes the inter-utterance quadruples, where terms come from different utterances. The "Avg" is the abbreviation for Average.**

| Datasets | ZH | EN |
|---|---|---|
| Dialogue | 1,000 | 1,000 |
| Utterance | 7,452 | 7,452 |
| Target | 8,308 | 8,264 |
| Aspect | 6,572 | 6,434 |
| Opinion | 7,051 | 6,933 |
| Quadruple | 5,742 | 5,514 |
| Intra | 4,467 | 4,287 |
| Inter | 1,275 | 1,227 |
| Avg. of #Word per dialogue | 206 | 181 |
| Avg. of #Utterance per dialogue | 7.45 | 7.45 |
| Avg. of #Quadruple per dialogue | 15 | 14 |

POS of the $a$-th token. We apply a MLP to predict the probability distribution of labels:

$$P(y_{a,b}^{sr}) = \text{softmax}(\text{MLP}(\mathbf{s}_{a,b}^{sr})), \qquad (13)$$

where the label $y_{a,b}^{sr} \in \{\text{T}, \text{A}, \text{O}, \text{None}\}$.

## 3.5 Training

For the MRM, the loss function is defined using the span pair-level cross-entropy loss:

$$\mathcal{L} = -\sum_{i=1}^{K} \sum_{j=i}^{K} y_{i,j} \log(P(y_{i,j})). \qquad (14)$$

For the Span Ranker, the loss function is defined using the span-level cross-entropy loss:

$$\mathcal{L}_{sr} = -\sum_{a=1}^{|I|} \sum_{b=a}^{|I|} y_{a,b}^{sr} \log(P(y_{a,b}^{sr})). \qquad (15)$$

## 4 EXPERIMENTS

## 4.1 Dataset and Metrics

We conducted experiments on the DiaASQ dataset [14] derived from posts and comments on the Chinese social media platform Weibo [6]. The dataset is in the mobile phone domain and includes Chinese (ZH) and English (EN) versions. Table 1 lists the statistics of the dataset.

Following previous works [14], we take identification-F1 (iden-F1 for short) [1] and micro-F1 as measurements for evaluating the DiaASQ task. The micro-F1 measures the whole quadruple, while iden-F1 does not distinguish the sentiment polarity. We also test the span match and pair extraction subtasks using exact-F1, where a prediction is only correct if the extracted span or pair matches the ground truth exactly. Besides, we use macro-F1 to measure the performance of Span Ranker.

---

[6]https://weibo.com

**Table 2: Main results on Chinese (ZH) and English (EN) datasets. The "T/A/O" is the abbreviation of Target/Aspect/Opinion. The "ICL" denotes in-context learning. The "/" means that models cannot execute the span match task. The "±" denotes the standard deviation of our results. The Micro-F1 in the DiaASQ task is our main measure.**

| | | Span Match (F1) | | | Pair Extraction (F1) | | | DiaASQ (Main) | |
|---|---|---|---|---|---|---|---|---|---|
| | | T | A | O | TA | TO | AO | Iden-F1 | **Micro-F1** |
| ZH | CRF-Extract-Classify | 91.11 | 75.24 | 50.06 | 32.47 | 26.78 | 18.90 | 9.25 | 8.81 |
| | SpERT | 90.69 | 76.81 | 54.06 | 38.05 | 31.28 | 21.89 | 14.19 | 13.00 |
| | ParaPhrase | / | / | / | 37.81 | 34.32 | 27.76 | 27.98 | 23.27 |
| | Span-ASTE | / | / | / | 44.13 | 34.46 | 32.21 | 30.85 | 27.42 |
| | BDTF-Dia | 91.08 | 76.24 | 60.88 | 51.41 | 49.33 | 52.58 | 41.06 | 34.22 |
| | GTS-Dia | 90.23 | 76.94 | 59.35 | 48.61 | 43.31 | 45.44 | 37.51 | 34.94 |
| | ChatGPT (5-shot ICL) | 68.78 | 57.87 | 36.45 | 34.98 | 42.48 | 27.43 | 20.59 | 18.41 |
| | Ours | **91.49**$_{\pm0.21}$ | **77.10**$_{\pm0.30}$ | **61.24**$_{\pm0.55}$ | **53.56**$_{\pm0.54}$ | **50.29**$_{\pm0.22}$ | **53.26**$_{\pm0.64}$ | **42.82**$_{\pm0.37}$ | **40.59**$_{\pm0.36}$ |
| EN | CRF-Extract-Classify | 88.31 | 71.71 | 47.90 | 34.31 | 20.94 | 19.21 | 12.80 | 11.59 |
| | SpERT | 87.82 | 74.65 | 54.17 | 28.33 | 21.39 | 23.64 | 13.38 | 13.07 |
| | ParaPhrase | / | / | / | 37.22 | 32.19 | 30.78 | 26.76 | 24.54 |
| | Span-ASTE | / | / | / | 42.19 | 30.44 | 45.90 | 28.34 | 26.99 |
| | BDTF-Dia | 88.60 | 73.37 | 62.53 | 49.26 | 47.55 | 49.95 | 38.80 | 31.81 |
| | GTS-Dia | 88.62 | 74.71 | 60.22 | 47.91 | 45.58 | 44.27 | 36.80 | 33.31 |
| | ChatGPT (5-shot ICL) | 68.05 | 53.22 | 45.08 | 28.76 | 37.24 | 25.36 | 17.17 | 15.26 |
| | Ours | **89.00**$_{\pm0.71}$ | **75.09**$_{\pm0.68}$ | **63.57**$_{\pm1.07}$ | **55.12**$_{\pm0.89}$ | **53.11**$_{\pm0.29}$ | **56.52**$_{\pm1.15}$ | **47.61**$_{\pm0.78}$ | **43.80**$_{\pm0.76}$ |

## 4.2 Baselines

Because DiaASQ is a new task, Li et al. [14] redesigned several existing methods and proposed a new token-pair tagging method, GTS-Dia, based on GTS [26]. We redesign another token-pair tagging method, BDTF-Dia, based on the table representation approach of the BDTF [34]. Besides, we utilize the few-shot in-context learning method to evaluate the ChatGPT [7]. The detailed settings for ChatGPT are in Appendix B. All baselines are as follows:

**Token-pair Tagging Methods**: GTS-Dia [14] and BDTF-Dia [34].
**Span-level Enumeration Method**: Span-ASTE [28].
**Few-shot In-Context Learning Method**: ChatGPT.
**Other Methods**: CRF-Extract-Classify [2], SpERT [9], and Para-Phrase [31].

## 4.3 Experimental Settings

All fine-tuned models use the Roberta-Large [21] and Chinese-Roberta-wwm-base [7] as PLMs for English and Chinese datasets. For our MRM, we set $K = 128$. The max length of the span is 10. We select the model with the highest micro-F1 scores on the validation set for the test set. Our experiments run on a single Nvidia RTX 3090 GPU with 24GB of memory, and all experimental results are the average values over five runs under the seed list [0, 1, 2, 3, 4].

## 4.4 Main Results

We compare our model against baselines in the DiaASQ task using micro-F1 and iden-F1 scores. Besides, we also evaluate the performance of our model in two sub-tasks (Span Match and Pair Extraction). Table 2 presents these results.

For the DiaASQ task, our model shows considerable and stable performance improvements, surpassing the previous state-of-the-art GTS-Dia by an average of **8.06%** in iden-F1 and **8.07%** in micro-F1 scores across two languages. We attribute these gains to our span-pair tagging scheme and the consideration of our model on the three characteristics of DiaASQ tasks mentioned in the Introduction. For the Pair Extraction task, our model achieves the best performance, reflected by exact-F1 scores exceeding 50% for TA, AO, and TO extraction on Chinese and English datasets. These improvements highlight the effectiveness of our model in capturing relations between terms through span-level interactions. Besides, the performance of Span-ASTE is unsatisfactory despite modeling span-level interactions. The reason is that the enumerated triplets are mostly invalid, leading to a severe label sparsity problem. The problem further prevents Span-ASTE from training toward the optimal solution. For the Span Match task, our model achieves an overall performance improvement in both languages. Furthermore, the performance of all models is poor in the Opinion Match task. The phenomenon is related to the diverse expressions of opinion.

Besides, we use 5-shot in-context learning to evaluate the performance of ChatGPT. On the one hand, ChatGPT performs better than CRF-Extract-Classify and SpERT in many cases, demonstrating its potential in resource-constrained scenarios. On the other hand, it still falls short compared to most PLMs-based fine-tuned models. The performance sharply decreases when dealing with complex tasks such as DiaASQ, indicating that ChatGPT struggles with understanding complex structured sentiment information. This conclusion is consistent with the observations of Zhang et al. [32] and Zhao et al. [37].

---

[7]https://chat.openai.com/

**Table 3: Ablation results.**

| Model | ZH | EN |
|---|---|---|
| MRM | **40.59** | **43.80** |
| *w/o* all modules | $36.80_{(\downarrow 3.79)}$ | $38.62_{(\downarrow 5.18)}$ |
| *w/o* $\mathbf{H}^{Sp}$ & $\mathbf{H}^{Re}$ | $39.35_{(\downarrow 1.24)}$ | $42.53_{(\downarrow 1.27)}$ |
| *w/o* $\mathbf{H}^{Sp}$ | $40.45_{(\downarrow 0.14)}$ | $42.89_{(\downarrow 0.91)}$ |
| *w/o* $\mathbf{H}^{Re}$ | $40.53_{(\downarrow 0.06)}$ | $43.00_{(\downarrow 0.80)}$ |
| *w/o* **wemb** | $40.41_{(\downarrow 0.18)}$ | $41.46_{(\downarrow 2.34)}$ |
| *w/o* **demb** | $40.08_{(\downarrow 0.51)}$ | $43.57_{(\downarrow 0.23)}$ |
| *w/o* $\mathbf{s}_i \odot \mathbf{s}_j$ | $38.71_{(\downarrow 1.88)}$ | $40.80_{(\downarrow 3.00)}$ |
| Span Ranker | **82.95** | **81.99** |
| *w/o* **pemb** | $82.62_{(\downarrow 0.33)}$ | $81.58_{(\downarrow 0.41)}$ |

**Table 4: Results on complex scenarios.**

| | Model | Overall | $\mathcal{D}_1$ | $\mathcal{D}_2$ | $\mathcal{D}_3$ |
|---|---|---|---|---|---|
| ZH | GTS-Dia | 34.94 | $30.19_{(\downarrow 4.75)}$ | $14.81_{(\downarrow 20.13)}$ | $25.29_{(\downarrow 9.65)}$ |
| | BDTF-Dia | 34.22 | $30.53_{(\downarrow 3.69)}$ | $32.98_{(\downarrow 1.24)}$ | $35.40_{(\uparrow 1.18)}$ |
| | Ours | 40.59 | $37.62_{(\downarrow 2.97)}$ | $42.34_{(\uparrow 1.75)}$ | $40.37_{(\downarrow 0.22)}$ |
| EN | GTS-Dia | 33.31 | $26.95_{(\downarrow 6.36)}$ | $31.25_{(\downarrow 2.06)}$ | $32.85_{(\downarrow 0.46)}$ |
| | BDTF-Dia | 31.81 | $30.00_{(\downarrow 1.81)}$ | $30.69_{(\downarrow 1.12)}$ | $30.89_{(\downarrow 0.92)}$ |
| | Ours | 43.80 | $43.09_{(\downarrow 0.71)}$ | $42.76_{(\downarrow 1.04)}$ | $42.97_{(\downarrow 0.83)}$ |

## 5 FURTHER ANALYSIS

### 5.1 Ablation Study

We conduct ablation studies to verify the effectiveness of different modules in MRM, using the micro-F1 scores in the DiaASQ task as the measure. Besides, we study the impact of POS information in the Span Ranker, using the macro-F1 scores in three span match subtasks as the measure.

As shown in Table 3, each module positively functions on MRM. Our model performs better than the state-of-the-art baseline even without all modules, demonstrating that MRM benefits from the span-pair tagging scheme. Specifically, the performance of MRM degrades on both Chinese and English datasets when removing speaker and reply attention modules $\mathbf{H}^{Sp}$ and $\mathbf{H}^{Re}$, demonstrating the necessity of dialogue structure information. The speaker and reply sequence can provide relevant information between utterances. Besides, MRM needs the width information of spans due to the imbalanced distribution of the width of terms, so removing the **wemb** hurts the performance. After removing the distance information between spans **demb**, our model cannot perceive the relative distance between spans in span pairs, so the performance drops. MRM can explicitly model the span-level interactions by Hadamard product $s_i \odot s_j$. Removing $s_i \odot s_j$ makes the performance of MRM sharply degrade. Finally, we study the effect of POS in Span Ranker. The performance improvement of the Span Ranker gains from POS information **pemb** is limited. One possible reason is that PLMs acquired substantial POS knowledge during pre-training.

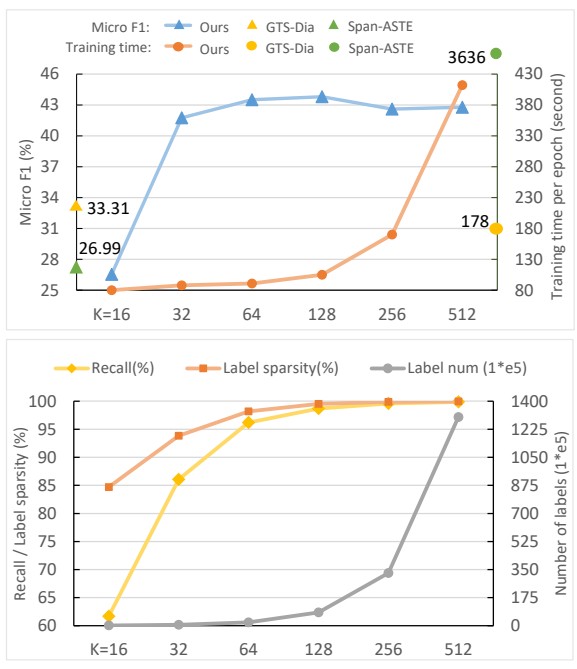

**Figure 5: Analysis results with respect to $K$ on the English dataset. In the second line chart, recall@$K$ denotes the ratio of the number of terms in the top $K$ spans to the total number of terms; Label sparsity@$K$ denotes the ratio of the number of None labels to the total number of labels in the $K \times K$ matrix; Label num@$K$ denotes the overall number of labels in the $K \times K$ matrix. The results of the Chinese dataset are in Appendix C.**

### 5.2 Detailed Study on Complex Scenarios

To verify the ability of our model to deal with complex scenarios, we conduct the three subsets of the test set: 1)$\mathcal{D}_1$: each dialogue in the $\mathcal{D}_1$ contains inter-utterance quadruples. 2)$\mathcal{D}_2$: the number of quadruples for each dialogue in the $\mathcal{D}_2$ is not less than 15. 3)$\mathcal{D}_3$: the number of utterances for each dialogue in the $\mathcal{D}_3$ is not less than 8. The $\mathcal{D}_1$, $\mathcal{D}_2$, and $\mathcal{D}_3$ accounted for 68%, 14%, and 48% of the ZH test set and 32%, 32%, and 48% of the EN test set. We compare the performance of our model with GTS-Dia and BDTF-Dia on the DiaASQ task, and the micro-F1 scores are in Table 4. The performance of GTS-Dia sharply drops when facing complex scenarios. BDTS-Dia performs more stably than GTS-Dia, but the performance is still unsatisfactory. In contrast, our model exhibits a stable and considerable performance when facing complex scenarios, outperforming the GTS-Dia and BDTF-Dia in all cases. The phenomenon suggests that our model is more effective in 1) capturing inter-utterance dependencies, 2) handling complex corresponding relations between terms caused by abundant quadruples, and 3) handling longer sequences.

### 5.3 Effect of $K$ value

The $K$ value denotes the number of remaining spans after pruning, determining the size of the span-level matrix. We study the effect

| | |
|---|---|
| **A:** 'OnePR 9Pro ?', **B:** 'The battery life is not good , plus the overheat [ eat melon ]', **C:** 'Have you used it ?', **D:** 'Holding the OnePlus 9P , the heat is really hot, and the power consumption is also very large, especially after updating the 12 system', **E:** "The battery life of the x3 is OK.", **F:** 'One plus has become an abandoned ship , it is better to buy the findx3 series'. | |
| Ground Truth / Ours: | {OnePR 9Pro, battery life, not good, negative}, {OnePlus 9P, heat, hot, negative}, {OnePlus 9P, power consumption, very large, negative}, {x3, battery life, OK, positive} |
| GTS-Dia: | {OnePR 9Pro, battery life, _good, positive}, {OnePlus 9P, heat, hot, negative}, {OnePlus 9P, _consumption, very large, negative}, {x3, battery life, OK, positive} ✗ ✗ |

**Figure 6: An example predicted by GTS-Dia and our model.**

**Table 5: Time Complexity. The $N$, $L$, and $K$ denote the number of tokens, spans, and remaining spans after pruning, where $L \gg N$ and $L \gg K$. We ignore the dimension of vector representations for convenience. It is worth noting that our span pruning strategy implemented by the Span Ranker differs from the span pruning strategy of the Span-ASTE.**

| Model | Time Complexity |
|---|---|
| GTS-Dia | $O(N^2)$ |
| Span-ASTE | $O(N^2) + O(K^3) = O(\max(N^2, K^3))$ |
| *w/o* span pruning | $O(N^2) + O(L^3) = O(L^3)$ |
| Ours | $O(N^2) + O(K^2) = O(\max(N^2, K^2))$ |
| *w/o* span pruning | $O(L^2)$ |

of the $K$ value on the English dataset, as depicted in Figure 5. We selected the model with the best performance in the validation dataset to evaluate the test dataset.

Specifically, the performance of our model is suboptimal with a small $K$ value, such as 16. The reason is that a small $K$ corresponds to a low recall, indicating that abundant terms do not appear in the $K \times K$ matrix. Subsequently, our model achieves state-of-the-art performance as $K$ increases due to increased recall and label number. However, the performance does not infinitely increase with the increase of $K$. The reasons for this phenomenon are as follows: 1) the high label sparsity makes it difficult for the model to converge to the optimal solution during training; 2) The number of non-None labels has an upper bound. Besides, the training time of our model positively correlates with the $K$ value. Our model exhibits extremely high training efficiency when $K$ is less than 256. In conclusion, our model demonstrates superior performance compared to GTS-Dia and Span-ASTE regarding effectiveness and efficiency, provided that a suitable value of $K$ is chosen (such as 64 and 128).

## 5.4 Analysis of Time Complexity

Table 5 shows the time complexity of the three models. On the one hand, our model outperforms the Span-ASTE with or without using the span pruning strategy because we construct the span-level matrix rather than enumerating all spans and triplets, which leads to Span-ASTE's $O(\max(N^2, K^3))$ time complexity. On the other hand, GTS-Dia has $O(N^2)$ time complexity due to its token-level matrix. The time complexity of our model is slightly worse

at $O(\max(N^2, K^2))$ because we enumerate spans and construct the span-level matrix. In practice, the time complexity of our model is optimized to $O(N^2)$ due to $K < N$. Besides, the Span Ranker accelerates the enumeration of spans in MRM by caching the span indices so that the running speed of our model is faster than GTS-Dia.

## 5.5 Case Study

In order to better understand the capacity of our model, we illustrate a case study using GTS-Dia and our model. As shown in Figure 6, our model can correctly extract all quadruples while GTS-Dia fails. Concretely, GTS-Dia ignores the *not* and wrongly identifies the sentiment polarity by tagging the token pairs $(OnePR, good)$ and $(9Pro, good)$ when extracting the first quadruple. In the third quadruple, GTS-Dia wrongly extracts *consumption* rather than *power consumption*. The reason is that GTS-Dia cannot capture the complete span-level semantics. In contrast, our model correctly extracts quadruples by tagging the span pairs.

## 6 CONCLUSIONS AND FUTURE WORK

This paper proposes a novel span-pair interaction and tagging method for the Dialogue-level Aspect-based Sentiment Quadruple analysis (DiaASQ) task, which includes a novel Span-pair Tagging Scheme (STS) and a simple and efficient Multi-level Representation Model (MRM). The STS explicitly captures the complete span-level semantics by tagging span pairs in a span-level matrix. MRM enumerates spans and constructs a span-level matrix of span pairs based on the STS, explicitly modeling the dialogue information and the span-level interactions. Besides, we train a Span Ranker to improve the running efficiency of the MRM. Extensive experiments on the DiaASQ datasets demonstrate the superiority of our method.

There are also several potential limitations in this work. When K is small, the Span Ranker may filter out some terms from the candidate span list with the result that they cannot appear in the span-level matrix. These errors further limit the performance of final quadruple decoding. In the future, we will extend our work and develop an end-to-end framework to solve this issue under the premise of efficiency. Besides, there is mutual supervision information between span-pair labels in the span-level matrix. We plan to apply contrastive learning to model label-level supervision information.

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

# A  QUADRUPLE DECODING

The quadruple decoding details are shown in Algorithm 1. Firstly, we extract all target, aspect, and opinion terms using the predicted label of span pairs only located in the matrix diagonal. Secondly, we extract all relations between terms and sentiment polarities using the predicted label of span pairs only located in the strictly upper triangular region of the matrix. Thirdly, we decode all *(target, aspect, opinion)* triplets by the extracted terms and *target-aspect* and *aspect-opinion* relations. Finally, we verify all triplets and obtain their sentiment polarities by the extracted *target-opinion* relations with sentiment polarities.

---

**Algorithm 1** Quadruple Decoding for DiaASQ

---

**Input:** Span-level tagging matrix $P = \{..., y_{i,j}, ...\}$ with length $K$, where $y_{i,j}$ denotes the predicted label; Label set {T, A, O, TA, AO, TO-POS, TO-NEG, TO-OTH}; The function I2S() denoting the mapping from Span Index to Span String.

**Output:** quadruples set $Q_2$

1: Initialize sets $\mathcal{T} = \{\}$, $\mathcal{A} = \{\}$, $O = \{\}$, $\mathcal{TA} = \{\}$, $\mathcal{AO} = \{\}$, $Q_1 = \{\}$, $Q_2 = \{\}$ and dict $\mathcal{TO} = \{\}$;

2: # 1. decoding term types from main diagonal region of $P$.

3: $\mathcal{T} = \{i \mid y_{i,i} = \text{T}, 0 \le i < K\}$

4: $\mathcal{A} = \{i \mid y_{i,i} = \text{A}, 0 \le i < K\}$

5: $O = \{i \mid y_{i,i} = \text{O}, 0 \le i < K\}$

6: # 2. decoding relations between terms and sentiment polarities from the strictly upper triangular region of $P$.

7: $\mathcal{TA} = \{(i, j) \mid y_{i,j} = \text{TA}, i \in \mathcal{T}, j \in \mathcal{A}, 0 \le i < j < K\} \cup \{(j, i) \mid y_{i,j} = \text{TA}, i \in \mathcal{A}, j \in \mathcal{T}, 0 \le i < j < K\}$

8: $\mathcal{AO} = \{(i, j) \mid y_{i,j} = \text{AO}, i \in \mathcal{A}, j \in O, 0 \le i < j < K\} \cup \{(j, i) \mid y_{i,j} = \text{AO}, i \in O, j \in \mathcal{A}, 0 \le i < j < K\}$

9: $\mathcal{TO} = \{\{(i, j) : *\} \mid y_{i,j} = \text{TO-*}, i \in \mathcal{T}, j \in O, 0 \le i < j < K, * \in \{\text{POS, NEG, OTH}\}\}$

10: $\mathcal{TO} = \mathcal{TO} \cup \{\{(j, i) : *\} \mid y_{i,j} = \text{TO-*}, i \in O, j \in \mathcal{T}, 0 \le i < j < K, * \in \{\text{POS, NEG, OTH}\}\}$

11: # 3. obtaining quadruples

12: $Q_1 = \{(a, b, c) \mid (a, b) \in \mathcal{TA},\ (b, c) \in \mathcal{AO}\}$

13: $Q_2 = \{(\text{I2S}(a), \text{I2S}(b), \text{I2S}(c), s) \mid (a, b, c) \in Q_1,\ (a, c) \in \mathcal{TO}.keys(),\ s = \mathcal{TO}[(a, c)]\}$

14: **return** $Q_2$

---

# B  PROMPT DESIGN FOR CHATGPT

We use the *gpt-3.5-turbo-16K* model of the OpenAI public API (version up to July 10) and design a prompt elaborately to test the performance on the DiaASQ task. The prompt (i.e., the input of ChatGPT) includes three parts:

1) **Instruction**. We use instruction to guide the ChatGPT what it needs to do. Our instruction is as follows:

*Given a conversation as input, you need to extract all (Target, Aspect, Opinion, Sentiment) quads.*

Meanwhile, we attach five annotations to the instructions so that the ChatGPT better understands the DiaASQ task. These annotations are summarized from the annotation rules of DiaASQ datasets [14], as shown in Table 6.

2) **Demonstrations**

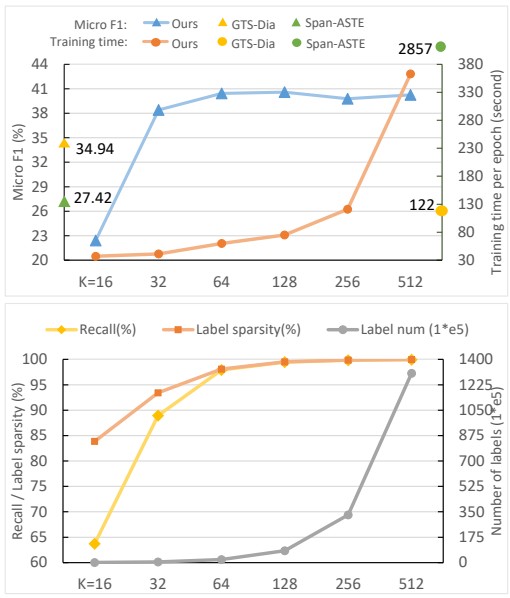

**Figure 7: Analysis results with respect to $K$ on the Chinese dataset.**

We achieve the few-shot in-context learning of ChatGPT by adding demonstrations. We use the 5-shot in-context learning due to the limitations of the input length. Each demonstration includes a dialogue as input and a two-dimension list as a quadruple set.

3) **Tested Sample** The tested sample is the sample ChatGPT need to test. Compared with the demonstration, the tested sample lacks the quadruple set.

The above three parts constitute the input of ChatGPT. We expect ChatGPT to output a two-dimensional list as a quadruple set of the tested sample. An example is shown in Table 6.

# C  EFFECT OF K VALUE ON THE CHINESE DATASET

As shown in Figure 7, the effect of K value on the Chinese dataset and English dataset is similar.

Received 20 February 2007; revised 12 March 2009; accepted 5 June 2009

**Table 6: An example of prompt for ChatGPT.**

| | | |
|---|---|---|
| Input | instruction | Given a conversation as input, you need to extract all (Target, Aspect, Opinion, Sentiment) quads. (1) Target represents the object to be described, such as mobile phone models, mobile phone brands, operating system, processor, etc. Besides, a demonstrative pronoun or general reference words can not be annotated as targets. For example, the word phone is not a target. (2) A noun phrase that indicates a specific aspect of the target should be annotated as a aspect. If multiple contiguous words indicate a fine-grained aspect, they should be annotated together as a whole aspect. For example, screen brightness instead of screen should be annotated as an aspect. Similarly, fast charge chip external frequency and screen size should be labeled as one aspect, respectively. (3) Words to express positive or negative or other emotional tendencies can be annotated as opinions, such as bad, wrong and good. (4) The opinions can reflect the sentiment orientation towards an aspect. The sentiment polarity of this task is divided into positive, negative , and other. In other words, The value range of Sentiment is [positive, negative, other]. (5) Target , Aspect and Opinion must be substring from the input. |
| | demonstrations | Input:
{'turn': 0, 'speaker': 0, 'utterance': 'Trust me , buying an old Apple is the best for your needs'}
{'turn': 1, 'speaker': 1, 'utterance': 'That charging speed , battery , signal can be used ?'}
{'turn': 2, 'speaker': 0, 'utterance': "Why ca n't it be used"}
{'turn': 3, 'speaker': 2, 'utterance': "As far as his cognitive level is concerned , do n't need Apple , Nokia is enough"}
{'turn': 4, 'speaker': 3, 'utterance': 'Signal is bad'}
{'turn': 5, 'speaker': 4, 'utterance': 'Battery ? Charge ?'}
Target-Aspect-Opinion-Sentiment Quads:
[['Apple', 'charging', 'can be used', 'other'], ['Apple', 'battery', 'can be used', 'other'], ['Apple', 'signal', 'can be used', 'other'], ['Apple', 'Signal', 'bad', 'negative']]
Input:
......
Target-Aspect-Opinion-Sentiment Quads:
...... |
| | tested sample | Input:
{'turn': 0, 'speaker': 0, 'utterance': 'Why sister Yi change back to Xiaomi ?'}
{'turn': 1, 'speaker': 1, 'utterance': 'find n ready to sell , no need for two phones'}
{'turn': 2, 'speaker': 2, 'utterance': 'Does the folding screen not good to use , sister Yi ?'}
{'turn': 3, 'speaker': 1, 'utterance': "It 's easy to use , but compared to the Xiaomi 11ultra flagship machine that is good in all aspects , some things are not good enough , such as taking pictures and charging, so I took the two mobile phones out together , and I found that I used Xiaomi more often ."}
{'turn': 4, 'speaker': 3, 'utterance': 'He has always used Xiaomi'}
{'turn': 5, 'speaker': 0, 'utterance': 'He changed to OPPO some time ago'}
Target-Aspect-Opinion-Sentiment Quads: |
| Output | quadruples | [['find n', 'taking pictures', 'not good', 'negative'], ['find n', 'charging', 'not good', 'negative'], ['Xiaomi 11ultra', 'taking pictures', 'not good', 'positive'], ['Xiaomi 11ultra', 'charging', 'not good', 'positive']] |

