# OpenReview forum: "Span-pair Interaction and Tagging for Dialogue-level Aspect-based Sentiment Quadruple Analysis"
_ACM.org/TheWebConf/2024/Conference — TheWebConf24 Oral_

### Official Review · Reviewer_yNtx · 2023-11-21

**Novelty:** 6
**Technical Quality:** 6

**Review:**

This paper introduces a novel span-based model tailored for the dialogue-level aspect-based sentiment analysis (DiaASQ) task.
Recognizing the limitations of previous method, which suffers rom incomplete span-level interaction and modeling, this study proposes a span-pair tagging schema as a solution.
Moreover, the span-based approach mitigates complexity through an efficient span pruning strategy.
Crucially, acknowledging the significance of dialogue-level features in this task, the paper employs a multi-level representation model.
This model facilitates comprehensive modeling of dialogue features and interactions between tokens and spans.
Experimental results demonstrate the model's efficacy, with an improvement exceeding 5% in F1 score across two benchmarks.
Subsequent experiments further validate the utility of the proposed module and confirm the model's superiority in complex scenarios.



**Pros**

1. **Advanced Span-Level Representation and Interaction**: The methods proposed in this paper leverage span-level representation and interaction, which holds significance for the DiaASQ task. The approach might be a notable advancement over prior token-based methods.

2. **Significant Performance Improvement**: The proposed methods have demonstrated a remarkable ability to surpass previous works in this task. This leap in performance signifies the effectiveness of the methods and also marks a testament to the innovative approach and robustness of the proposed model.

3. **Efficient Pruning for Time Efficiency**: The model incorporates an effective pruning method, enhancing its time efficiency significantly. This aspect is crucial for practical applications where processing speed is as important as accuracy. The ability to maintain high performance while also being time-efficient makes the model more applicable in real-world scenarios, where vast amounts of data require quick processing.

4. **Well-Written and Clear Methodology**: The paper is well-written, with a particularly clear and easy-to-follow methods section. This clarity in writing is essential for the dissemination of content and encourages further research and application of the proposed methods.



**Cons**

1. **Inconsistencies in Time Complexity Description**: There are some contradictions in the paper regarding the time complexity of GTS-Dia. The paper cites different complexity orders in different sections, leading to confusion about the actual computational efficiency of the model. This discrepancy needs clarification to fully understand the model's scalability and applicability.

2. **Lack of Clear Distinction from Previous Works**: The paper does not fully characterize how its methods differ from previous works, such as Span-ASTE. A more detailed comparison or a clearer delineation of the unique aspects of the proposed methods would enhance the reader's understanding of the paper's contributions and the advancements it brings to the field.

**Questions:**

1. **Clarification on Time Complexity**: There appears to be a contradiction in the stated time complexity of GTS-Dia. Specifically, line 157 cites a time complexity of $O(L^3)$, whereas line 834 mentions $O(N^2)$. However, GTS-Dia seems to enumerate the triplets of (target, aspect, opinion) only for valid terms and solely during the test phase. This approach unlikely results in a high $O(L^3)$ complexity in training stage. Could you clarify the actual time complexity, particularly considering that the model only enumerates triplets for valid terms during the test phase and not during training?

2. **Decoding Algorithm Query**: Regarding the decoding algorithm presented in Algorithm 1, it seems that a triplet is formed only if the target-aspect and aspect-opinion pairs are valid. The polarity of the triplet is determined by the target-opinion pair. The question arises why, in the initial stage, the algorithm does not verify the validity of the target-opinion pair. Additionally, if target-opinion is not a valid pair, how is the sentiment polarity of the entire triplet determined?

3. **Comparison with Span-ASTE**: The author notes that the pruning strategy in this paper differs from that in Span-ASTE. Beyond the different pruning strategies, are there any other significant distinctions between the proposed methods and Span-ASTE? Understanding these differences could highlight the unique contributions of your work.

4. **Typographical Error**: There is a typo error on line 153 – an extra period is present. Correcting this would improve the paper's overall readability and professionalism.

**Reviewer Confidence:**

4: The reviewer is certain that the evaluation is correct and very familiar with the relevant literature

**Scope:**

4: The work is relevant to the Web and to the track, and is of broad interest to the community

---

### Official Review · Reviewer_2hLL · 2023-11-22

**Novelty:** 4
**Technical Quality:** 4

**Review:**

This work focuses on the recent research task, Dialogue-level Aspect-based Sentiment Quadruple analysis (DiaASQ) and introduces a method with a span-pair tagging scheme (STS) and a multi-level representation model (MRM). The main contribution can be its span tagging accuracy when compared to the selected baselines. However, apart from the validated effectiveness and the extensive ablation studies that show the effectiveness of different components, there are multiple questions that require clarification.  These questions are detailed in the next part of this review.

**Questions:**

Here are the questions that require clarification:

1. The quadruple decoding process is not clearly discussed in the paper itself, which is not self-contained. Specifically, even adding the discussions in Appendix A, it remains unclear how the filtering of invalid triplets and identification of sentiment polarity are addressed, which is a core component of the proposed method.

2. Some conclusions or arguments are placed arbitrarily and without sufficient justifications. For example, line 378 of this paper says, "The labels are distributed across different regions of the span-level matrix in the STS, which reduce the difficulty of tagging." However, it remains unclear which method this difficulty is compared to and shows the reduced difficulty of tagging. Later, "In contrast, in the GTS-Dia scheme, all labels will likely appear in the strictly upper triangular region of the token-level matrix, making tagging challenging." this discussion is not only difficult to justify the correctness and also leads to confusion: is the STS tagging not having the labels in the upper triangular region? So, a clear description is required to follow the contributions.

3. The notations are overly used and sometimes repeated without proper illustration. In the third point of section 3.2.3, "h2h" and "t2t" labels are mentioned without proper introduction, then the notation 'h' is further used to denote the contextual representations and H and later again used in Equation (5). Hence, it is not easy to comprehend the part of information or variable these notations refer to. In addition, in line 487, the $demb_l$ is introduced to denote the learnable embeddings of distance $l$. However, it is not clearly discussed about how the corresponding embeddings are computed.  In summary, by having these many ambiguous descriptions of the proposed methods, it is challenging to grasp the contributions of this work and identify the flaws to improve as well.

**Ethics Review Description:**

No.

**Reviewer Confidence:**

3: The reviewer is confident but not certain that the evaluation is correct

**Scope:**

3: The work is somewhat relevant to the Web and to the track, and is of narrow interest to a sub-community

---

### Official Review · Reviewer_qPag · 2023-11-25

**Novelty:** 2
**Technical Quality:** 6

**Review:**

### pros:
* The author's method optimized the previous method in terms of asymptotic time complexity, increasing it from O (K ^ 3) to O (K ^ 2), which is a significant optimization.
* The author designed a pruning method (span ranker), which controlled the number of span pairs and further improved the efficiency of the algorithm.
* The author's method and experimental description are very detailed and clear to understand.
* The author compared its method with a large number of baselines. The comparison with large models such as ChatGPT is very convincing.

### cons:
* In daily dialogues, you can find coreferences and ellipses , meaning there are often omitted parts in utterances. In the method proposed by the author, directly performing Span-pair Tagging and ellipses on the spans in the sentence is a problem that needs to be solved. The use of Incomplete Utterance Rewriting (IUR) is a feasible solution.
* The author has reduced the time complexity of the step of generating span pairs to be tagged in previous work from O (K ^ 3) to O (K ^ 2), but the matrix to be filled is still very sparse, and many spans do not have practical significance. One possible optimization is to first perform entity extraction and recognition, combining syntactic structures, reducing the number of candidate spans to O (N).
* In the case study in section 5.5, only one baseline was compared. Providing more cases would be more intuitive.
* In the face of long dialogues, the author's O (K ^ 2) algorithm is still not efficient enough, and the number of span pairs will rapidly increase.
* This method lacks novelty and only reduces the number of candidate span pairs by designing algorithms and pruning.

**Questions:**

* What is the comparison between the author's method and baseline in terms of specific runtime? The author can provide specific runtime to better demonstrate how much algorithm efficiency has improved.
* What is the specific non-null rate of the matrix of span pairs? Please provide it to demonstrate how sparse it is.
* What is the case study of how large models such as ChatGPT generate specific results? I am curious about what kind of result will be given.

**Reviewer Confidence:**

3: The reviewer is confident but not certain that the evaluation is correct

**Scope:**

2: The connection to the Web is incidental, e.g., use of Web data or API

---

### Official Review · Reviewer_5G8X · 2023-11-28

**Novelty:** 4
**Technical Quality:** 5

**Review:**

This paper proposes a new method for dialog-level aspect-based sentiment extraction. Their method is built on a previous approach from the DiaASQ task that uses token-level interaction matrix to extract the aspects, opinions and sentiments. Their approach improves the previous method by considering the span-level interactions, which consists of 3 modules: span generating/pruning, span-pair generating/pruning and decoding. With the pruning technique, the efficiency of their approach can be dramatically improved. They evaluated their method on the DiaASQ task and show good improvements against previous methods. The ablation studies and example help understand how their method works.

Pros:
- A simple and practical approach for dialog-level aspect-based sentiment extraction.
- Good effectiveness gains compared to previous methods.
- Efficiency improvements.
- The paper is clearly written and easy to follow.

Cons:
- The evaluations are only done on a single benchmark. Proving its effectiveness across various benchmark (might be hard to find applicable dataset though) is good to demonstrate its generalizability.
- The baseline could be further improved. For example, adding a baseline on finetuning a LLM to do the aspect extraction would be great to understand the necessity of special design for this task.

**Questions:**

n/a

**Reviewer Confidence:**

3: The reviewer is confident but not certain that the evaluation is correct

**Scope:**

3: The work is somewhat relevant to the Web and to the track, and is of narrow interest to a sub-community

---

### Decision · Program_Chairs · 2024-01-22

**Decision:**

Accept (Oral)

**Comment:**

I see no reason that this paper should not be accepted. While, various concerns were raised in the initial round of reviewers, the authors have discussed these concerns in detail in their responses and have promised various updates and corrections to address them. Even in the absence of these promises, I believe the paper has sufficient merit to recommend acceptance. In addition, one reviewer had identified themselves as a champion of the work.

 I recommend oral presentation below because I believe the technical quality and novelty are sufficient to merit an oral presentation, although perhaps the topic is a little narrow. I am not calibrated on the breakpoint between oral and poster presentation. The paper does not urgently demand oral presentation, and if space does not permit it, a poster presentation would be fine.